# Cost of surgical treatment for ulnar nerve entrapment in Finland, 2011–2015: a registry-based cost description study

Aarni Hannula ![ORCID],[1] Laura Miettinen,[1] Kaisa Lampainen,[1] Jorma Ryhänen,[1] Paulus Torkki ![ORCID],[2] Sina Hulkkonen[1]

[1]Department of Hand Surgery, Helsinki University Hospital and University of Helsinki, Helsinki, Finland
[2]Department of Public Health, Helsingin Yliopisto, Helsinki, Finland

**Correspondence to**
Dr Aarni Hannula;
aarni.hannula@helsinki.fi

## ABSTRACT

**Objectives** The aim of this study was to evaluate the cost of surgical treatment for primary ulnar nerve entrapment (UNE) borne by the public sector in Finland.

**Design** Registry-based cost description study.

**Setting** Primary and secondary care throughout Finland.

**Participants** We identified all the patients diagnosed with primary UNE in the whole population of Finland from 2011 to 2015 from the Care Register for Health Care. From these patients, we identified those who had undergone ulnar nerve release during the year they were diagnosed or the following year.

**Interventions** Open ulnar nerve release.

**Outcome measures** The primary outcome measure was cost borne by the public sector in 2015 euros. The cost of surgery was based on the diagnosis-related group prices. We calculated the cost of a single visit to a primary care physician, an electroneuromyography examination, a preoperative visit to a hand surgeon and a follow-up appointment by telephone in specialised care for each patient. These unit costs were provided by the Finnish Institute for Health and Welfare and the same costs were used for each patient. We obtained the number of reimbursed sick days and the total amount reimbursed to each patient in euros within the 2 years after diagnosis from the Social Insurance Institution of Finland.

**Results** During our study period, approximately 1786 primary UNE diagnoses were made yearly, and on average, 876 (49%) of patients received surgical treatment annually. The surgery-related cost per patient averaged at EUR 1341 (43%) and reimbursed sick leaves at EUR 952 (30%) during this period. The annual cost of surgical treatment for UNE borne by the public sector in Finland varied between EUR 3082 and EUR 3213 per patient.

**Conclusions** The average cost of surgical treatment for UNE in Finland was EUR 3140 per patient between 2011 and 2015.

## INTRODUCTION

The ageing population and rising healthcare costs are increasing the need to prioritise efficient treatment strategies. To enable an objective comparison of different treatment strategies, the costs must be estimated and the health-related results of interventions must be obtained. Compression neuropathies of the upper limbs are common conditions and

---

**STRENGTHS AND LIMITATIONS OF THIS STUDY**

⇒ The data from the Finnish Care Register for Health Care and the Social Insurance Institution of Finland, both covering the whole population of Finland, made it possible to accurately estimate the cost of reimbursed sick leave paid by the public sector.

⇒ The treatment protocol for ulnar nerve entrapment is uniform throughout Finland.

⇒ We did not compare the costs or efficacy of different medical interventions, and our results cannot be used directly in healthcare decision-making.

⇒ The number and types of healthcare contacts, in addition to the actual surgery and routine electroneuromyography, was not based on exact research material but on expert opinion of routine practice in Finland.

---

are often surgically treated. However, studies examining the costs of ulnar nerve entrapment (UNE) or its surgical treatment are scarce. The aim of this study was to evaluate the costs of surgical treatment for UNE borne by the public sector in Finland between 2011 and 2015.

### Ulnar nerve entrapment

UNE is the second most common compression neuropathy in the upper extremity.[1] The most common compression site is at the elbow, at the cubital tunnel, whereas compression in the wrist in Guyon's canal is much less frequent.[2] A Finnish population-based study concluded that the average age-standardised incidence rate of UNE per 100 000 person-years was 36.1 among men and 26.3 among women.[3] UNE most commonly affects working-aged people and its incidence peaks around late middle age in both sexes.[3–6] UNE can cause numbness, paraesthesia and pain in the arm and hand following ulnar nerve distribution and weakness of the ulnar nerve-innervated muscles. In Finland, diagnosis is based on typical

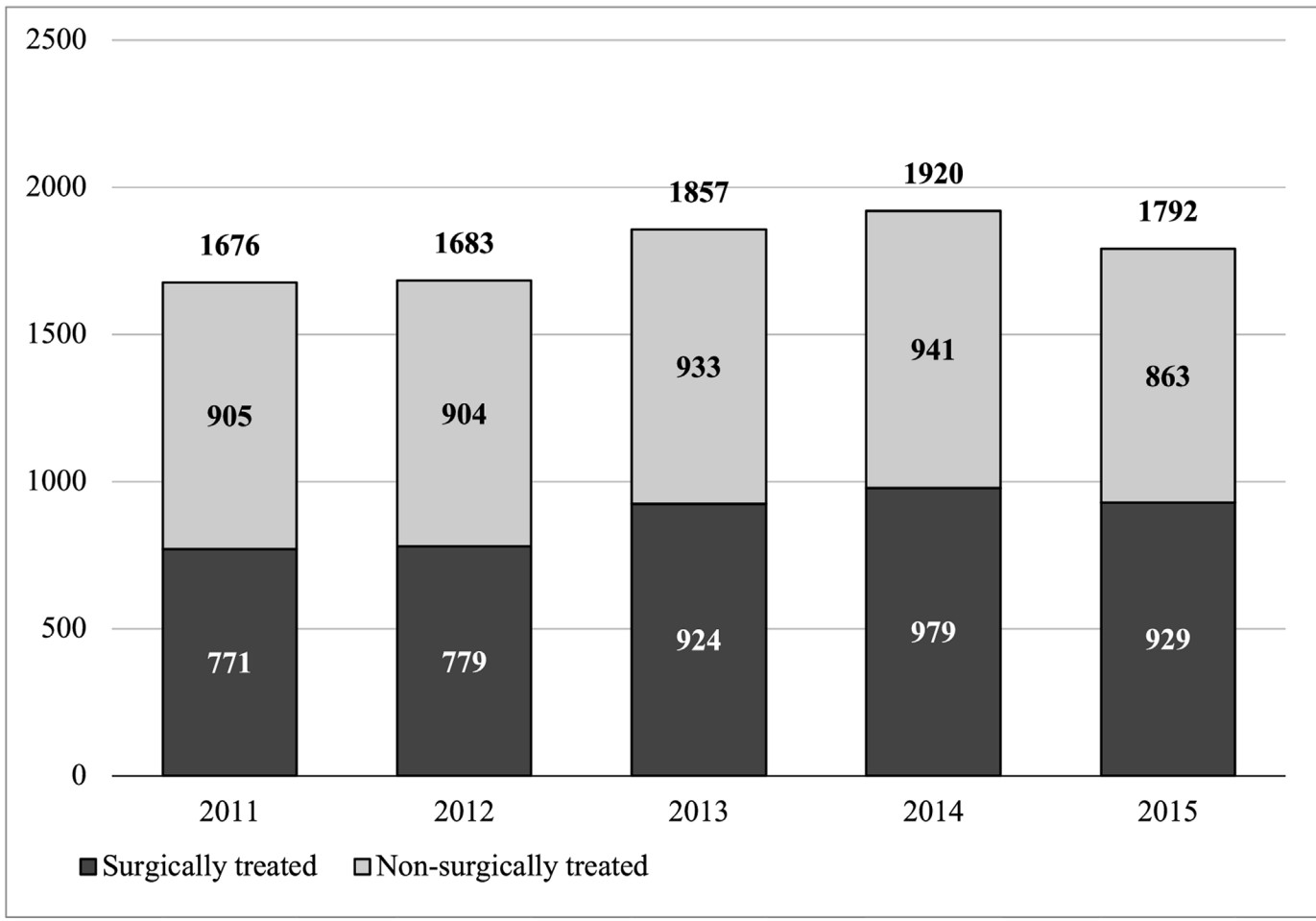

**Figure 1** Number of surgically and non-surgically treated patients.

symptoms, clinical findings and positive electroneuromyography (ENMG) findings.

## Surgical treatment of UNE

Most patients with mild symptoms benefit from conservative treatment and can be treated without surgery.[2] Prevalent surgical techniques for cubital tunnel syndrome include local decompression and anterior subcutaneous or submuscular transposition of the ulnar nerve. Medial epicondylectomy can be incorporated if needed. The effectiveness of anterior transpositions and of simple decompression does not seem to differ.[1]

In Finland, the prevalent surgical technique is simple decompression with surgical block anaesthesia, and the procedure is most often performed as day surgery. The endoscopic technique is exceedingly rare in Finland.

Complications are relatively rare. A meta-analysis estimated an average complication rate of 3% and revision rate of 2% after cubital tunnel release among all different surgical techniques. Simple decompression appears to be the safest option.[7] In a 2017 retrospective cohort study, simple decompression had a 3.8% (N=6) complication rate, of which 1.3% (N=2) cases required surgery.[8]

## METHODS
### Study registries

In Finland, every public and private healthcare provider is obligated by law to provide information on procedures and inpatient episodes to the Finnish Institute for Health and Welfare (FIHW) for administrative and research purposes. Patient-level data, including social security number, diagnoses and the Nordic classification of surgical procedure codes, are stored in the Care Register for Health Care—a national register of both public and private hospital data in Finland.[9] The diagnoses are coded according to the International Classification of Diagnoses (ICD). The code for UNE diagnosis is in accordance with the tenth revision of ICD in 1996.

The Social Insurance Institution of Finland (Kela) keeps a register of the total amount paid in euros and of the incidence of benefits and their distribution by region and population group. This comprises all social benefits, including reimbursed sick leaves and drugs. Kela compensates 70% of workers' average pay after 10 days of sick leave, for up to 300 days.[10] We obtained permission from the FIHW and Kela to study and publish from these registries.

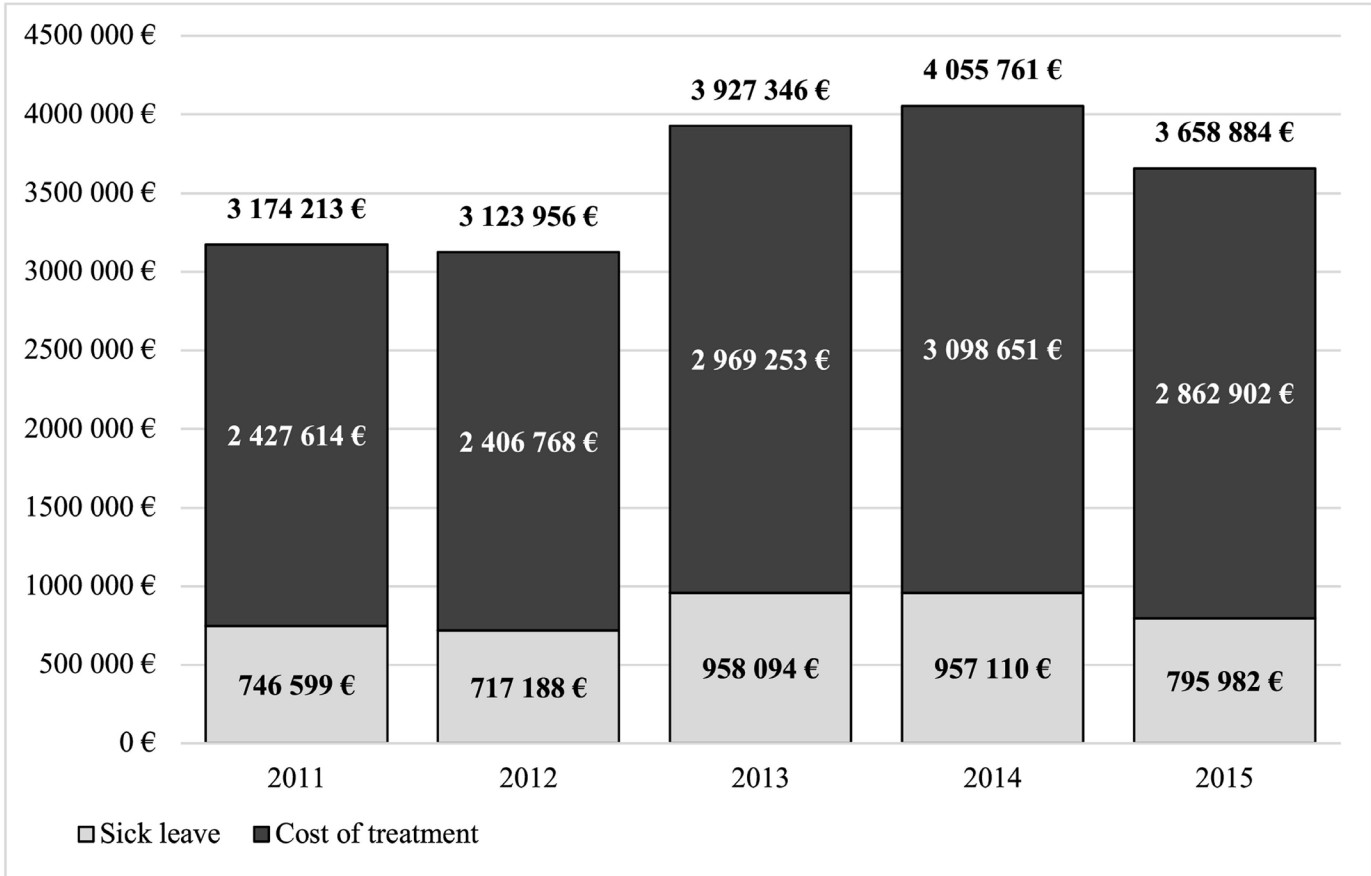

**Figure 2** Total cost of surgically treated patients in Finland in 2015 euros.

## Population

The study population was the whole population of Finland between 2011 and 2015. The Care Register for Health Care provided the total number of patients diagnosed with UNE during the study period. We included those with the diagnosis code G56.2 according to the tenth revision of ICD. Diagnosis code G56 (mononeuropathies of the upper limb) was also defined as UNE if the cases had a concomitant G56.2 diagnosis. Only incident cases were included in the study evaluating the costs of primary UNE. All patients with a concomitant diagnosis of carpal tunnel syndrome were excluded. Surgically treated patients were defined as those with the NSCP code ACC53, ulnar nerve release, which had occurred during the year of diagnosis or the following year.

## Cost framework

To approximate the cost of the surgical treatment of UNE, we combined the costs of diagnosis and referral in primary healthcare, ENMG examination, outpatient visit and telephone follow-up in specialised care, surgery and its related costs and the amount of euros reimbursed for sick leaves. All costs are expressed in 2015 euros, using the consumer price index.[11]

## Cost of surgery and reimbursed sick leaves

As the cost of surgery, we used the diagnosis-related group (DRG) prices from 2011 and 2017 provided by the FIHW.

These prices are based on the financial data from all five university hospitals and five central hospitals in Finland. The prices for the missing years of 2012–2015 were linearly interpolated from these years. The DRG prices comprised the total cost of care, including overheads but not patient fees. We validated these prices by conducting a survey on DRG prices among the financial services of 10 hospital districts that used DRG pricing during the study period. To estimate the total cost of surgery based on our survey, we calculated the population-weighted averages from the DRG prices.

The DRG008O cost code (surgery of nervous system, other, short treatment, non-complicated) was used to estimate the operating costs of cubital tunnel release. This included day surgery nerve operations, excluding carpal tunnel release, which has its own DRG code.

The total amount of funds paid for sick leaves with a primary diagnosis of UNE (with the same exclusions and inclusions as mentioned above) and the total amount of reimbursed days of sick leave and their distribution by length (exceeding 10days) were acquired from Kela's register.

## Approximated costs

For diagnosis and referral in primary care, we counted one outpatient visit with a general practitioner. One ENMG examination, used to diagnose entrapment neuropathies

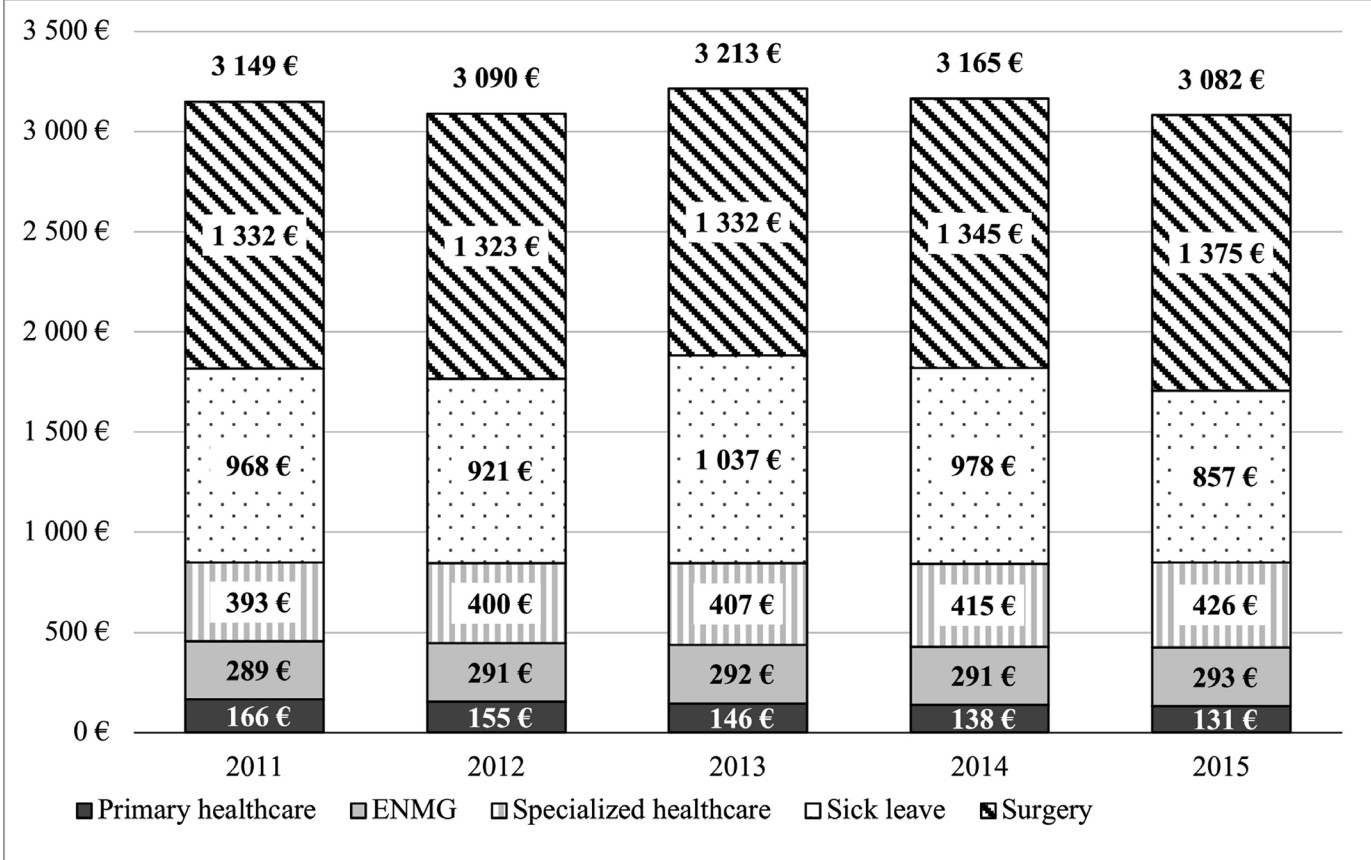

**Figure 3** Cost components per patient in 2015 euros. ENMG, electroneuromyography.

was also calculated per patient. We counted one preoperative surgeon appointment and one follow-up telephone appointment in specialised care as this was the most usual arrangement in our experience. A visit to a nurse in primary care was counted as a cost for the removal of stitches.

The unit cost for an outpatient appointment with a general practitioner and a surgeon, and an outpatient appointment with a nurse in primary care was provided by the FIHW. For the unit cost of the follow-up appointment by telephone and the ENMG examination, we used the unit costs of the largest hospital district in Finland and its diagnostic centre, as the FIHW did not provide these unit costs.

## ETHICAL APPROVAL
We obtained permission from the FIHW and the Social Insurance Institution of Finland to study and publish from these registries.

## Patient and public involvement
None.

## RESULTS
During our study period, 1786 (SD 106.9) new UNE diagnoses were made on average per year. Of the patients diagnosed with UNE, 876 (49%, SD 95) underwent surgical treatment on average per year. Figure 1 shows the number of patients with newly diagnosed UNE who underwent surgery or received non-surgical treatment per year.

Of the patients diagnosed with UNE, 5057 (56.6%) were men and 3871 (43.4%) women. The mean age was 50.84 (SD 14.95). Of the patients who underwent surgery, 2558 (58.4%) were men and 1824 were women (41.6%). Their mean age was 51.77 (SD 13.99).

Figure 2 shows the total annual cost of the surgical treatment of patients. During the years under study, the total cost per 100 000 population was EUR 57.567 to EUR 74.122. The cost of treatment varied between EUR 44,351 (76% of total costs) and EUR 56,630 (78%) per 100 000 people, and sick leave between EUR 13 216 (22%) and EUR 17 576 (24%) per 100 000 population. Of the surgically treated patients, an average of 328 (37%) had more than 10 sick leave days annually. The average length of these sick leaves was 40 days. This resulted in an average of 13 000 reimbursed days of work annually, incurring an average total cost of EUR 840 000 per year and an average cost of EUR 952 per surgically treated patient.

Figure 3 shows the cost formation per year. The results of our validation survey are shown as supplementary material (online supplemental table 1). There was no statistical difference between the mean of the annual DRG prices in our survey and the interpolated prices provided by FIHW (t-test, p=0.42).

## DISCUSSION

We approximated the amount of public funds used in the treatment chain to surgically treat one patient with UNE as EUR 3082–3213 (2015 currency) in Finland in 2011–2015. The cost of treatment did not rise during the study period.

Our study's strength is that it included the cost of primary healthcare and a nerve conduction examination, which are often excluded or left unclear. In addition, our study population represented the whole of the Finnish population, not only a subgroup, such as Medicare beneficiaries. The register covers the entire Finnish population and has been shown to be from satisfactory to very good in terms of completeness and overall accuracy, though this has not been studied exclusively in compression neuropathies.[9] UNE diagnostics are reasonably uniform in Finland, as an ENMG is required for diagnosis. In Finland, every citizen is entitled to sick leave with pay, healthcare services and surgical treatment when needed. This ensures patients' livelihoods and prevents them returning to work prematurely.

The study also has limitations. It only examined one upper extremity and did not consider possible revision procedures. Moreover, we used the DRG008O price for day surgery. The cost associated with possible hospital stay in the ward after surgery is not reflected in our results. In our experience, patients are seldom monitored in the ward overnight. It is also possible that the sick leaves in this study include bilaterally operated patients' sick leaves, that is, the other extremity was also operated on within 2 years of initial surgery. This would inflate the cost of reimbursements. The same is true for sick leaves resulting from the treatment of complications and possible revisions if the same ICD-10-code was used to prescribe the sick leaves. Sick leaves lasting 10 days or less were missing, as they are not registered by Kela. This sick leave period does not represent the cost from our perspective, but is nonetheless a cost to the employer, who is mandated to pay the employee's salary for this period.

Using a time trade-off survey and a decision analytic model, a 2012 study estimated that simple decompression, including possible complications and revisions, offers 2.76 quality-adjusted health years (QALY) of health benefit per treated patient in comparison to no treatment.[12] This is a substantial health benefit, considering the magnitude of the costs involved and the commonly used cost-effectiveness thresholds for acceptable cost of QALY in the reimbursement of pharmaceutics, for example, GPD 20 000–30 000 by the National Institute of Health and Care Excellence in the UK.[13] In our study, we could not measure health benefits from the registries we used. A full economic evaluation, including both costs and benefits of all the relevant treatment strategies, could be a useful topic for future research.

Most often, postoperative pain is treated with a short regimen of non-steroidal anti-inflammatory drugs, and as the standard reimbursement for a pack of one hundred 400 mg ibuprofen tablets is under five euros, we excluded the cost of drug reimbursements as insignificant. In Finland, opioids are not commonly used for managing post-operative pain after minor surgical procedures such as the standard procedure for UNE.

We did not estimate the cost of complications, as they are not expected to significantly increase overall costs. Neither did we assess the contribution and cost impact of subsequent revisions and bilateral surgeries in this study, as they could not be analysed on the basis of the data.

The cost of UNE and its treatment has previously mostly been examined in studies comparing the costs associated with different surgical techniques, as it seems that these are equally effective.[1 12 14 15] The direct surgical cost of the open decompression of the cubital tunnel, based on Medicaid reimbursements in 2005–2012 was US\$ 878–1149,[16] which is slightly lower than our DRG price. However, in the same study, the hospital charges were many times higher than the Medicaid reimbursements, indicating a possible shortfall.

A cost analysis conducted as part of a randomised controlled trial in the Netherlands calculated that 94% of costs were caused by loss of production, including non-paid work, which was monetarily valued. It reported a very low cost of treatment—only EUR 151 per patient.[15]

In conclusion, we estimated that the cost of the surgical treatment chain of UNE was on average EUR 3140 in Finland in 2011–2015. This amount is similar to that of previously made estimates in the USA.

**Contributors** AH and LM contributed equally to this research. JR, PT and SH planned the study. AH, LM, KL, JR, PT and SH made a substantial contribution to the design of the work. AH, LM, KL and SH contributed to the acquisition and analysis of the data. AH and LM drafted the article. AH, LM, KL, JR, PT and SH revised the work for intellectual content. AH submitted the study. AH, LM, KL, JR, PT and SH approved the article for publication. AH, LM, KL, JR, PT and SH agreed to be accountable for all aspects of the work in ensuring that questions related to the accuracy or integrity of any part of the work are appropriately investigated and resolved. S.H. is the guarantor for the study.

**Funding** The authors disclose receipt of the following financial support for the research, authorship and/or publication of this article: This work was supported by the Finnish Medical Foundation grant for SH, grant number 5302. Open access was funded by Helsinki University Library.

**Competing interests** The authors disclose receipt of the following financial support for the research, authorship and/or publication of this article: In the past 36 months, PT has received grants or contracts from Merck, AstraZeneca, Roche, Abbvie and Promedical; consulting fees from Nordlic Healthcare Ltd and has acted in a leadership or fiduciary role in the Effectiveness Society (Vaikuttavuusseura ry) and the Association for children and youths with disabilities, Promedical Ltd. All other authors declare no competing interests.

**Patient and public involvement** Patients and/or the public were not involved in the design, or conduct, or reporting, or dissemination plans of this research.

**Patient consent for publication** Not applicable.

**Ethics approval** The study was approved by the Ethics Committee of Northern Ostrobothnia (ETTMK 107/2017). Informed consent was not sought for the present study because this is a register study, and according to Finnish law, no informed consent is mandatory.

**Provenance and peer review** Not commissioned; externally peer reviewed.

**Data availability statement** Data may be obtained from a third party and are not publicly available. The research permit granted by the Finnish Institute for Health and Welfare and The Social Insurance Institution of Finland to publish from their

records does not allow sharing the data to external parties. Acquiring the same data requires Finndata approval, instructions may be found at https://findata.fi/en/.

**ORCID iDs**
Aarni Hannula http://orcid.org/0000-0002-2067-9933
Paulus Torkki http://orcid.org/0000-0002-1127-4205

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
