## [Reviewer comments · BMJ Open]

ARTICLE DETAILS

TITLE (PROVISIONAL)	Cost of surgical treatment for ulnar nerve entrapment in Finland, 2011–2015: a registry-based cost description study
AUTHORS	Hannula, Aarni; Miettinen, Laura; Lampainen, Kaisa; Ryhänen, Jorma; Torkki, Paulus; Hulkkonen, Sina

VERSION 1 – REVIEW

REVIEWER	Power, Dominic Queen Elizabeth Hospital Birmingham, The Birmingham Peripheral Nerve Injury Service
REVIEW RETURNED	23-Dec-2022

GENERAL COMMENTS	I would specifically mention how revision data is collected and handled and if cannot be tracked then state that this may include revision cases which could affect the return to work and costs. The authors have deliberately avoided the analysis of treatment costs for complications and treatment failure and this is mentioned. The literature review should include more up to date references eg Wade R Network Systematic Review 2020 JAMA Networkopen that details the risk of different procedure. It would be helpful to know whether there is endoscopic surgery performed and whether this is coded differently / has a different cost. Some commentary on measuring health outcomes to look at the QALYs of the procedure would be helpful. Although not available in this dataset it would enhance the discussion of study limitations, promote the need for a core outcome dataset and the need for full Health economic analysis of the procedure and benefits rather than the cost of care delivery in isolation. Overall an interesting read limited by the lack of data granularity but with large numbers and a national review. Well done on your project - a few minor revisions are suggested.
--

REVIEWER	Sarmiento, Samuel Johns Hopkins University School of Medicine, Plastic and Reconstructive Surgery
REVIEW RETURNED	03-Feb-2023

GENERAL COMMENTS	Good, informative article overall, even though the scope is limited. That said, this cost description can serve as a comparator for other health systems in various health economics contexts. I know that costs descriptions are not easy, which is why it's important to have a reference from health systems where the entire population is included.
---

	Here are my recommendations:  1. The article definitely needs language adjustments throughout. This should start with the title, where the range should read "from 2011 to 2015." The authors might want to consider English editing services if someone proficient in English is not available at their institution. 2. I know the article focuses on costs for a specific surgical procedure. However, it would be highly informative for readers to know the demographics of the population included in the study. If these data are available to you, I would highly recommend you include it in the study. This is important because if another health economics study uses this as a reference, demographic characteristics such as age are important to consider for study design purposes. 3. If you're under the word limit for the abstract, please be more descriptive. For example, under "Primary and secondary outcomes," be more detailed about what cost is being measured, what it includes, excludes, etc. Many people will just read your abstract, so you want to be as clear, precise, and compelling as possible. Make it count.
--	--

VERSION 1 – AUTHOR RESPONSE

Reviewer: 1

Mr. Dominic Power, Queen Elizabeth Hospital Birmingham

Comments to the Author:

I would specifically mention how revision data is collected and handled and if cannot be tracked then state that this may include revision cases which could affect the return to work and costs.

Thank you, it is indeed possible that our estimate for cost of sick leave could include sick leaves after revisions if ICD-code G56.2 was used by the physician to prescribe the sick leave. We have added a clarifying statement to the discussion section on page 11.

The authors have deliberately avoided the analysis of treatment costs for complications and treatment failure and this is mentioned.

Thank you, we have tried to be as transparent as possible.

The literature review should include more up to date references eg Wade R Network Systematic Review 2020 JAMA Network Open that details the risk of different procedure.

Thank you, we have added 2 references regarding the risk of complications and revisions on page 6.

It would be helpful to know whether there is endoscopic surgery performed and whether this is coded differently / has a different cost.

Thank you for the well-founded remark. Endoscopic surgery is coded with the same procedural code and does not have separate diagnosis-related group price. Our wording on the use of endoscopic

technique was imprecise. Endoscopic cubital tunnel release is very rare in Finland, and we would argue so rare, that it has no significant effect on our results. We have changed the wording regarding the commonly used technique in the introduction section on page 6.

Some commentary on measuring health outcomes to look at the QALYs of the procedure would be helpful. Although not available in this dataset it would enhance the discussion of study limitations, promote the need for a core outcome dataset and the need for full Health economic analysis of the procedure and benefits rather than the cost of care delivery in isolation.

Thank you. It is true that our study is limited in providing direct assistance in healthcare decision making as we are measuring only the cost of only a single treatment option. We have added a paragraph in the discussion section promoting this shortcoming and discussing potential health benefits on page 12.

Overall an interesting read limited by the lack of data granularity but with large numbers and a national review.

Well done on your project - a few minor revisions are suggested.

Thank you!

Reviewer: 2

Dr. Samuel Sarmiento, Johns Hopkins University School of Medicine

Comments to the Author:

Good, informative article overall, even though the scope is limited. That said, this cost description can serve as a comparator for other health systems in various health economics contexts. I know that costs descriptions are not easy, which is why it's important to have a reference from health systems where the entire population is included.

Thank you!

Here are my recommendations:

1. The article definitely needs language adjustments throughout. This should start with the title, where the range should read "from 2011 to 2015." The authors might want to consider English editing services if someone proficient in English is not available at their institution.

Thank you, the manuscript has now been checked and corrected by our university's language editing services.

2. I know the article focuses on costs for a specific surgical procedure. However, it would be highly informative for readers to know the demographics of the population included in the study. If these data are available to you, I would highly recommend you include it in the study. This is important because if another health economics study uses this as a reference, demographic characteristics such as age are important to consider for study design purposes.

Thank you, a very good remark. We have added the demographic characteristics of the population in the results section on page 10.

3. If you're under the word limit for the abstract, please be more descriptive. For example, under "Primary and secondary outcomes," be more detailed about what cost is being measured, what it includes, excludes, etc. Many people will just read your abstract, so you want to be as clear, precise, and compelling as possible. Make it count.

Thank you, we have updated the abstract in terms of precision and length.

VERSION 2 – REVIEW

REVIEWER	Sarmiento, Samuel Johns Hopkins University School of Medicine, Plastic and Reconstructive Surgery
REVIEW RETURNED	03-Mar-2023
GENERAL COMMENTS	As per my previous request, please change title to "...from 2011 to 2015..." Thank you for addressing all the reviewers' comments. After making this change, I recommend the manuscript for publication.